**An efficient approach for inverting rock exhumation from thermochronologic age-elevation**
**relationship**
Yuntao Tian[1,2*], Lili Pan[1], Guihong Zhang[1], Xinbo Yao[1]
[1] Guangdong Provincial Key Laboratory of Geodynamics and Geohazards, School of Earth
Sciences and Engineering, Sun Yat-sen University, Guangzhou 510275, China
[2] Southern Marine Science and Engineering Guangdong Laboratory (Zhuhai), Zhuhai 519082,
China
*Corresponding author:
Yuntao Tian
tianyuntao@mail.sysu.edu.cn

**Abstract**

This study implements the least-squares inversion method for solving the exhumation history from thermochornologic age-elevation relationship (AER) based on the linear equation among exhumation rate, age and total exhumation from the closure depth to the Earth surface. Modelling experiments suggest significant and systematic influence of initial geothermal model, the *a priori* exhumation rate and the time interval length on the *a posterior* exhumation history. Lessons learned from the experiments include that (i) the modern geothermal gradient can be used for constraining the initial geothermal model, (ii) a relatively higher *a priori* exhumation rate would lead to systematically lower *a posteriori* exhumation, and *vice versa*, (iii) the variance of the *a priori* exhumation rate controls the variation of the inverted exhumation history, (iv) the choice of time interval length should be optimized for resolving the potential temporal changes in exhumation. Putting together these findings, we implemented a new stepwise inverse modeling method for optimizing the model parameters by comparing the observed and predicted thermochronologic data and modern geothermal gradient to mitigate the model dependencies on the initial parameters. Finally, method demonstration was performed using four synthetic datasets and three natural examples of different exhumation rates and histories. It is shown that the inverted rock exhumation histories from the synthetic datasets match the whole picture of the "truth", although the temporal changes in the magnitude exhumation are underestimated. Modelling of the datasets from natural samples produce geologically reasonable exhumation histories. The code and data used in this work is available in GitHub (https://github.com/yuntao-github/A2E_app).

**Key words:** Thermochronology; Exhumation; Numerical inversion; Age-elevation relationship; Least-squares method; Geothermal model

## 1. Introduction

Quantifying rock exhumation from the Earth interior to the surface is important information for better understanding many geological problems, ranging from orogenic growth (e.g., Zeitler et al., 2001; Whipp Jr. et al., 2007) and decay (e.g., House et al., 2001; Hu et al., 2006), to resource and hydrocarbon evaluation and exploration (e.g., Armstrong, 2005; Mcinnes et al., 2005), as well as the underpinning endogenic and exogenic processes and their interactions (e.g., Burbank et al., 2003; Fox et al., 2015; Tian et al., 2015). Various experimental and modeling methods have been invented for estimating the rock exhumation at different crustal levels (e.g., Braun, 2003; Reiners and Brandon, 2006; Anderson et al., 2008; Braun et al., 2012; Fox et al., 2014).

One type of the methods for estimating the rock exhumation in the middle and upper crust relies on thermochronologic cooling ages acquired from by noble gas and fission-track dating of a series of accessory minerals, such as Ar-Ar, fission-track and (U-Th)/He analyses (Ault et al., 2019 and references therein). Based on the closure temperature theory (Dodson, 1973), assuming monotonic cooling, a thermochronologic age records the time duration that a rock cooled through the corresponding closure temperature, which is a function of the kinematics describing fission-track annealing and noble gas diffusion, and rock cooling rate (Dodson, 1973). If the depth of the closure temperature isotherm can be estimated from the crustal temperature field, a time-averaged exhumation rate can be obtained from the cooling age.

Based on the thermochronologic methods and thermo-exhumation modelling, many analytical and numerical tools have been implemented for inverting the exhumation and/or the associated cooling history from thermochronologic data. These tools have different functions, such as inverting temperature history (Laslett et al., 1987; Ketcham, 2005; Gallagher, 2012), determining time-averaged exhumation rates (Brandon et al., 1998; Ehlers, 2005; Willett and

Brandon, 2013; Glotzbach et al., 2015; Van Der Beek and Schildgen, 2023), spatiotemporal
changes in exhumation (Sutherland et al., 2009; Herman et al., 2013; Fox et al., 2014; Willett et
al., 2020), and evolution of exhumation in two or three dimensions given a tectonic framework
(Batt and Brandon, 2002; Braun, 2003; Van Der Beek et al., 2010; Valla et al., 2011; Braun et al.,

2012).

Convincing estimate of exhumation history for a region requires both a proper sampling
strategy for thermochronologic data and a robust modeling approach for exhumation inversion,
especially when the rock exhumation and its spatiotemporal changes are tectonically controlled
(Ehlers and Farley, 2003; Schildgen et al., 2018). A routine and efficient sampling strategy
acquires themochronologic ages from an elevation transect over a significant relief and a relatively
confined spatial distance. Plotting the age versus elevation, i.e., the age-elevation relationship
(AER), and analyzing the slope changes of the plot can provide first-order understanding of the
exhumation history (Fitzgerald et al., 1986). Because both the subsurface geothermal field and
closure temperature of thermochronometers are functions of the thermal advection and cooling
during rock exhumation (e.g., Dodson, 1973; Brandon et al., 1998), as well as the long-wavelength
topography (Braun, 2002; Ehlers and Farley, 2003; Glotzbach et al., 2015), Estimating reliable
exhumation rates requires to account for temporal variations of the thermal field caused by changes
in the thermal and kinematic boundary conditions.
Fox et al. (2014) reported a linear inversion modeling method that solves exhumation
history from AER, given a combination of *a priori* exhumation rates and assumed geothermal
parameters. However, as shown in that study, the inverted exhumation history depends highly on
these *a priori* values and geothermal assumptions. Building on that study, we here provide a
detailed test on the method and report an improved modeling method that makes use of both the
AER and the modern geothermal gradient for inverting exhumation history.

**2. Linear inversion method**

Our inversion of exhumation from thermochronologic data followed the linear inversion

approach of Fox et al. (2014). Rock Exhumation from the closure depth of a thermochronometer,
$z_c$, to the Earth's surface can be described as an integral of the exhumation ($\dot{e}$) from the cooling
age ($\tau$) to the present (Brandon et al., 1998; Fox et al., 2014). For a set of correlated bedrock
samples with a shared history of exhumation rates ($\dot{\mathbf{e}}$), their thermochronologic ages (**A**) and the
corresponding closure depths ($\mathbf{z_c}$) can be expressed by the following equation.

$$\int_0^\tau \dot{e}\, dt = z_c \quad \Rightarrow \quad \mathbf{A}\dot{\mathbf{e}} = \mathbf{z_c} , \tag{1}$$

where **A** is a model matrix, with n rows (the total number of samples) and m columns (the total
number of time intervals). Each row of the matrix is a discretization of a sample age, which is
composed of a number of time lengths ($\Delta t$) followed by an age residual ($R_i$) and a number of zeros.
The $\dot{\mathbf{e}}$ is a m-length vector of exhumation rates, and the $\mathbf{z_c}$ is n-length vector of closure depths.

This linear equation can be solved using the Least-Squares Regression approach assuming

the Gaussian uncertainties and *a priori* mean exhumation rate ($\dot{\mathbf{e}}_{pr}$) and associated variance $(\sigma_{pr})$
(Tarantola, 2005; Fox et al., 2014). Such an approach requires a m*m-sized parameter covariance
matrix, **C**, and a n*n-sized data covariance matrix, $\mathbf{C_\varepsilon}$, which includes the uncertainties on the
closure depths. These two matrices can be constructed as equations 2 and 3, respectively.

$$C_{ij} = \begin{cases} \sigma_{pr}^2, & if\ i = j \\ 0, & if\ i \neq j \end{cases} \tag{2}$$

$$(C_\epsilon)_{ij} = \begin{cases} \dot{e}_{pr}\epsilon_i\ , & if\ i = j \\ 0, & if\ i \neq j \end{cases}, \tag{3}$$

where $\dot{e}_{pr}$ and $\sigma_{pr}$ are the *a priori* exhumation and the associated variance, and the $\varepsilon_i$ is analytical
uncertainty of the age data. The construction of the data covariance matrix assumes the age data
are uncorrelated. Worth noting is that previous studies used different constructions of the data
covariance, changing from using the analytical age uncertainties (Fox et al., 2014; Fox et al., 2015)
to constant values (Jiao et al., 2017; Stalder et al., 2020).
Given the above model parameters, the equation 1 has a maximum likelihood solution for
the exhumation rate vector:

$$\dot{\mathbf{e}}_{po} = \dot{\mathbf{e}}_{pr} + \mathbf{C}\mathbf{A}^T(\mathbf{A}\mathbf{C}\mathbf{A}^T + \mathbf{C}_\epsilon)^{-1}(\mathbf{z}_c - \mathbf{A}\dot{\mathbf{e}}_{pr}),\qquad(4)$$

where $\dot{\mathbf{e}}_{pr}$ is a n-length vector of $\dot{e}_{pr}$, $\mathbf{z}_c$ is the n-length vector of closure depths calculated using a
combination of exhumation and geothermal model parameters (see section 3). The $\dot{\mathbf{e}}_{po}$ is the
posteriori maximum likelihood estimate of the exhumation rate, with a covariance matrix, $\mathbf{C}_{po}$,
which provides an estimate of the uncertainties on the model parameters (equation 5).

$$\mathbf{C}_{po} = \mathbf{C} - \mathbf{C}\mathbf{A}^T(\mathbf{A}\mathbf{C}\mathbf{A}^T + \mathbf{C}_\epsilon)^{-1}\mathbf{A}\mathbf{C}\qquad(5)$$

The method also provides a model resolution matrix, $\mathbf{R}$, which gives a measure on how
well the model estimates correspond to the true values:

$$\mathbf{R} = \mathbf{C}\mathbf{A}^T(\mathbf{A}\mathbf{C}\mathbf{A}^T + \mathbf{C}_\epsilon)^{-1}\mathbf{A}\qquad(6)$$


## 3. Closure depth and topographic correction

Inversion of the exhumation using the equation 1 requires accurate estimates of the closure
depths of the thermochronologic ages ($\mathbf{z_c}$), i.e., the depth of the closure temperatures (Fig. 1). The
latter can be determined by modelling the temperature of the crust using a 1D thermal-kinematic
model, which accounts for heat conduction, advection and production (Turcotte and Schubert,

2002):

$$\frac{\partial T_m}{\partial t} = \kappa \frac{\partial^2 T_m}{\partial z^2} + \dot{e} \frac{\partial T_m}{\partial z} + A_b, \tag{7}$$

where $A_b$ is the heat production (in °C/Myr). This function can be numerically solved using a
Crank–Nicolson time integration with a set of initial and boundary conditions, such as an initial
geothermal gradient (G0) at the start time of the model and surface temperature ($T_S$) (Turcotte and
Schubert, 2002; Fox et al., 2014).
The closure temperature ($T_c$) of a thermochronometer is a function of cooling rate ($\dot{T}$) at
the closure time and kinetic parameters of Helium and Argon diffusion and fission-track annealing
in mineral phases (Dodson, 1973):
$$\dot{T} = \frac{\Omega R T_c^2}{E_a} \exp\left(\frac{-E_a}{RT_c}\right), \tag{8}$$

where $\Omega$ and $E_a$ are the diffusion frequency factor normalized by the mineral size and geometry,
and activation energy, respectively. Parameter $R$ is the gas law constant. See reviews by Reiners
and Brandon (2006) for the $\Omega$ and $E_a$ parameter values for different thermochronometers.
The cooling rate ($\dot{T}$) can be computed from the derivative of transient geotherms, $T_m(t,z)$
that can be computed using equation 7 (Fox et al., 2014):
$$\dot{T} = \frac{\partial T_m}{\partial t} + \dot{e} \frac{\partial T_m}{\partial z}, \tag{9}$$

where $\dot{e}$ is unknown exhumation that can be computed through the equation 1.
Combining the equations 7-9, the closure depth of a thermochronological system ($z_{c,m}$) can
be numerically computed. This depth also needs a topographic correction, because of the
topographic perturbation, $p$, on the isotherms (Braun, 2002; Ehlers and Farley, 2003; Fox et al.,
2014; Glotzbach et al., 2015). Such a perturbation can be determined by the following equation
(Mancktelow and Grasemann, 1997; Fox et al., 2014):
$$p(\lambda) = \left(\frac{\gamma_0 - \gamma_a}{\gamma_{z_m}}\right) \exp\left(-z_m \left(\frac{\dot{e}}{2\kappa} + \sqrt{\left(\frac{\dot{e}}{2\kappa}\right)^2 + (2\pi\kappa)^2}\right) h(\lambda), \tag{10}\right.$$

where $\gamma_a$ is the atmospheric lapse rate, $\gamma_0$ and $\gamma_{z_m}$ are the thermal gradients at the model surface and
at the depth $z_m$. The $h(\lambda)$ is a cosine function expression of the model surface topography, which
can be determined using the discrete Fast Fourier Transform at the frequency domain. Here we use
the SRTM30 data for computing the topography of regions of interests.

Finally, the closure depth of the $z_c$ is corrected by the topographic perturbation (e.g.,

Brandon et al., 1998):

$$(z_c)_i = (z_{c,m})_i - p_i + h_i, \tag{11}$$

where $z_{c,m}$ is the closure depth calculated using the 1D geothermal model, $p$ and $h$ are the
topographic perturbation and elevation difference with respect to the mean elevation at the sample
site (Fig. 1), and the $i$ denotes the $i$-th age.

As shown by the equations 7, 8 and 9, the closure depth is a non-linear function of rock

cooling and exhumation. Therefore, the problem of interest is non-linear, which can be addressed
by iterative numerical modelling methods. In this work, the solution of exhumation is
approximated by coupling and iterating the linear inversion and closure depth modeling. As shown
in Tarantola (2005) and Fox et al. (2014), the algorithm converges in a few iterations and produces
stable outputs.

**4. Model evaluation**

Quantitative model assessment relies on a misfit value, i.e., the difference between

observed and predicted ages weighted by the observed analytical uncertainty:

$$\Phi_\tau = \sqrt{\frac{1}{N} \sum_{i=1}^{N} \left( \frac{\tau_{prd,i} - \tau_{obs,i}}{\varepsilon_i} \right)^2}, \tag{12}$$

where $\tau_{obs,i}$ and $\tau_{prd,i}$ are the observed and predicted $i$-th age calculated from the exhumation history,
and $\varepsilon_i$ is the uncertainty of the observed $i$-th age. Following Fox et al. (2014), both the *a priori* and
*a posteriori* misfits, $\Phi_{\tau,\,pr}$ and $\Phi_{\tau,\,po}$, are determined for the models. The difference between these
two misfit values provides a measure of the model improvements. A smaller posteriori misfit value
indicates an improved model result, and *vice versa*.
To evaluate the geothermal parameters, we also determined the misfit value of the
predicted to the observed modern geothermal gradient value using the following equation:
$$\Phi_{\gamma} = \sqrt{\left(\frac{\gamma_{prd} - \gamma_{obs}}{\varepsilon_{\gamma}}\right)^2},\qquad(13)$$

where $\gamma_{prd}$ and $\gamma_{obs}$ are the predicted and observed geothermal gradients, and $\varepsilon_{\gamma}$ is the uncertainty
of the observed value. Because the depth-temperature curves are slightly non-linear, the predicted
geothermal gradient ($\gamma_{prd}$) is calculated as a mean value for the upper 1 km of the model. Similar
as the assessment of age data, we also determined the *a priori* and *a posteriori* misfits, $\Phi_{\gamma,\,pr}$ and
$\Phi_{\gamma,\,po}$ values for assessing the geothermal parameters.

**5. The reference inverse model**
Following Willett and Brandon (2013) and Fox et al. (2014), here we use the published
AFT data acquired from Denali Massif (Fitzgerald et al., 1995) for method demonstration (Fig.
2a). A break-in-slope is shown by the AER at ~7-6 Ma, indicating a coeval change in slope, i.e.,
the apparent exhumation rate (Fitzgerald et al., 1995), increasing from $0.17 \pm 0.04$ km/Myr to 1.2
$\pm 0.6$ km/Myr (Fig. 2b). AER regression of young dates from the lower part of the transect
(between 4.3-2.0 km) also predicts a closure depth that is the intercept at $-3.3 \pm 3.4$ km (Fig. 2b).
However, using the present geothermal gradient (38.9 °C/km) (Fox et al., 2014) and a nominal
closure temperature of AFT method (110 °C) (Reiners and Brandon, 2006) and a -12 °C surface
temperature (Fox et al., 2014), the closure depth is predicted as ~3.1 km beneath the mean elevation
(~4 km), which is equivalent to an elevation of ~0.9 km. This closure depth is significantly higher
than the intercept (-3.3 ± 3.4 km). Such a difference indicates the AER slope of the lower part
overestimates the exhumation rates since ~7-6 Ma.
Following the protocol outlined in Fox et al. (2014), the reference inverse model uses the
following parameters, a start time at 25 Ma, a time interval ($\Delta t$) of 2.5 Myr, a 4020 m mean
elevation, a -12 °C surface temperature, *a priori* exhumation rate of 0.5 ± 0.15 km/Myr, a 24 °C/km
initial geothermal gradient, a 38.9 °C/km present geothermal gradient, a model block with a
thickness of 80 km, and a 30 km$^2$/Myr thermal diffusivity.
The exhumation history output of the reference model is shown in Fig. 3. The inversion
results reveal an more than two-fold increase of exhumation rate to a value of ~0.6 km/Myr at 7.5
Ma (Fig. 3b), consistent with the development of the break-in-slope in the AER. The model also
shows a gradual decrease of exhumation rate from *a priori* exhumation rate (0.5 km/Myr) to 0.3
km/Myr from 25 Ma to 7.5 Ma. The invariant exhumation during the starting stage resulted from
the fact that all ages are younger than 17.5 Ma, and thus the data have no resolution for the time
span. These results are similar to those of Fox et al. (2014). The posteriori misfit for the age is
1.88, significantly smaller than that of the priori model (4.51), suggesting the improvement by the
inverse modeling (Fig. 3b). Such a model also provides reasonable fit to the modern temperature
field, as shown by the small misfit (0.39) in the geothermal gradient (Fig. 3b).
The resolution of the inverted exhumation history can be assessed by the resolution matrix
**R** (equation 6). Imaging of the matrix shows the model provides no resolution for the time period
before 17.5 Ma (Fig. 3c), consistent with the fact that the oldest input age is younger than 16.1 ±
0.9 Ma. For the time span between 15 and 5 Ma, the model resolution is high, as shown by the
diagonal elements of the matrix, with the highest resolution at 7.5-5 Ma span, including eight age
date points (Fig. 3c). The most recent two phases of exhumation (5-0 Ma) are less resolved, as
shown by the nearly equal resolution values for the two phases, i.e., the latest four pixels of the
matrix (Fig. 3c). This is because no input ages fall into this time span, when the modeled
exhumation results are time-averaged values. The slight decrease in the last stage reflects changes
in geothermal gradient.

For assessing the correlation among model parameters, the calculated covariance matrix is

scaled by the diagonal covariance matrix (Fox et al., 2014):
$$\hat{C}_{\xi\beta} = \frac{C_{\xi\beta}}{\sqrt{C_{\xi\xi}}\sqrt{C_{\beta\beta}}}$$    (14)

The correlation matrix for the reference model is shown in Fig. 3d. The diagonal correlation

values are 1 and off-diagonal ones are dominantly negative, indicating anti-correlated uncertainties
(Fig. 3d), which suggests exhumation parameters were not resolved independently by the modeling.
In fact, it is expected to have the anti-correlation, because, given two steps of rock exhumation,
decreasing the exhumation during one step would increase that of the other step.

**6. Dependence on model parameters and proposed solutions**

Here we use the Denali data set for demonstrating the influences of (1) the initial

geothermal parameters, (2 and 3) the *a priori* mean and variance values of the exhumation rates,
and (4) time interval length on the inverted exhumation history. Also discussed in this section are
the solutions for optimizing the model setup for these parameters.

**6.1. Dependence on initial thermal model**

Different initial model geothermal parameters would lead isotherms to shift either

downward to greater depths or upwards to the Earth surface, and either compression or expansion
among isotherms. Therefore, the initial thermal models have systematic influence on the closure
depths and consequently the *a posterior* exhumation.

This is demonstrated by modelling experiments presented in Figure 4. Using a relatively

lower initial geothermal gradient produces relatively higher *a posterior* exhumation rates
(comparing the models shown in Figs. 4a-4f), and *vice versa*. Such an influence is significant even
for the time and elevation intervals with multiple age constraints (10-5.0 Ma). For example, using
relatively lower geothermal gradients of <22 °C/km would yield significantly higher average
exhumation rates of >0.75 km/Myr for the last two stages (<5 Ma) (Figs. 4a-4c) than those (<0.6
km/Myr) using higher initial geothermal gradients of ≥26 °C/km (Figs. 4d-f). Further, it is also
shown that models using higher and lower prior geothermal gradients of <20 °C/km (Figs. 4a-4b)
and >30 °C/km (Figs. 4e-4f) yield worse misfits ($\Phi_{\gamma, po}$ > 1) for the observed present-day
geothermal gradient than those ($\Phi_{\gamma, po}$ < 1) using medium initial gradients (22-26 °C/km) (Figs. 3
and 4c-4d).

These results highlight the importance of taking geothermal parameters into account in

inverting the exhumation history and model evaluation. We proposed to run a set of models using
different *a priori* geothermal parameters, especially the initial geothermal gradient, to search for
the proper intitial geothermal setup that provides reasonable fits to both the ages and the modern
geothermal gradient (see section 7 for details).

**6.2. Dependence on the *a priori* exhumation rate**

Both the mean and variance of the *a priori* exhumation rate have important influences on

the model solution for the maximum likelihood estimation method. Our modeling experiments
show that the mean value of the *a priori* exhumation has systematic influences on the inverted
exhumation. Similar to the reference model, exhumation of the preceding three stages (25-17.5
Ma) without age constraints is the same as the *a priori* input. For the following stages, a relatively
higher mean value of the *a priori* exhumation results in relatively lower *a posterior*i exhumation
rates (comparing different models presented in Fig. 5). For example, models using the mean *a*
*priori* exhumation of ≤0.4 km/Myr yield *a posterior* exhumation of 0.5-0.9 km/Myr for the stages
<7.5 Ma (Figs. 5a-5c), whereas those using a higher *a priori* value (≥ 0.6 km/Myr) result in *a*
*posterior* exhumation of 0.45-0.6 km/Myr for the same stages (Figs. 5d-5f). This is because a
relatively higher *a priori* value, which would be used for calculating thermal models, would lead
to a quicker increase in geothermal gradient and thus relatively shallower closure depths and
relatively lower exhumation rates.

The variance of the *a priori* exhumation rate has important influence on both the

exhumation rates and the posterior variance. Models with lower *a priori* variances yield less
variations in the *a posterior* exhumation history, and *vice versa* (comparing models in Fig. 6).
Further, models using the input variance of the *a priori* exhumation of 0.2-0.3 km/Myr (40-60%
of the mean value), the variation of the inverted exhumation history becomes stable (Figs. 3, 6c-
6d). Given that the uncertainty of the input age data, which is often 10%-20% at a two-sigma level,
larger variance of the inverted exhumation would be unreasonable (Figs. 6e-6f), especially when
multiple age data are available at different elevations.

We proposed to run a set of models using different *a priori* mean value of erosion rates to

search for the one that provides appropriate fits to both the ages and the modern geothermal
gradient. As to the *a priori* variance, we propose to use a value 30-70% of the *a priori* erosion rate.
Future applications of the method may need to test a set of the variance inputs so as to get a stable
exhumation output. Larger *a priori* variance would lead to larger uncertainties for the exhumation
rates, which is unreasonable and non-meaning for geological studies.

**6.3. Dependence on time interval length**

Constraining the onset time of major changes in exhumation rates is one of the important
tasks for inverting the exhumation history from thermochronologic data. Using a large time
interval length cannot accurately capture the potential transition time of exhumation rates. As
shown in the Figs. 7b-7d, models using time lengths of ≤3.5 Ma show an abrupt increase in
exhumation at 7-6 Ma, consistent with that shown in AER plot. However, the models using a large
time interval length (≥4.5 Ma) overestimate the onset time of the enhanced exhumation (Figs. 7e-
7f). Further, a relatively shorter time length would smooth temporal changes in exhumation rates,
leading to an underestimating of the variations. For example, as shown in the Fig. 7a, the model
using a relatively shorter time length (0.5 Ma) yields an exhumation variation between 0.35-0.60
km/Myr, significantly lower than those using relatively larger time interval lengths (Figs. 7b-7f).
In addition, a shorter time length also significantly increases the computational time and resources,
especially when processing a large number of vertical transects.
Given the interests in major exhumation changes, we propose the time interval length ($\Delta t$)
should be optimized for constraining the transitional time and the associated exhumation changes.
Therefore, the time interval length should be set as the absolute uncertainty at two sigma levels at
the break point ($\tau_b$) (equation 15). If the break point is unclear in AER, we suggest to use the
absolute uncertainty at two-three sigma levels at the median age value ($\tilde{\tau}$) (equation 15), so as to
focus on the time intervals where ages cluster.
$$\Delta\tau = \begin{cases} \delta\tau_b, & \textit{if a break in slope exists} \\ \delta\tilde{\tau}, & \textit{if no clear break in AER} \end{cases}, \tag{15}$$

where $\delta$ is the relative age uncertainty at two sigma levels, varying between 10%-20% among
different studies. Following this method, the Denali case should use a time length of ~1.5 Ma (7
Ma × 20%), slightly lower than that used in the reference model (Fig. 3).

**7. A new modeling guideline**

Following the modelling protocol outlined above, a stepwise modeling guideline is

developed for addressing the model dependencies on the initial geothermal parameter, the *a priori*
exhumation rates and time interval length. As illustrated in the Figure 8, the approach includes the
following three steps.

(i) Estimating a time-averaged erosion rate. Dividing each nominal closure depth, which

can be estimated from the nominal closure temperatures and the modern geothermal gradient, by
the corresponding age results in a time-averaged erosion rate. Then, a mean value can be
determined by averaging the rates. Such a mean value and assumed variance (30% - 50% in this
work) will be used as the *a priori* erosion rate.

(ii) Optimizing the fit to the modern geothermal gradient. This step runs a set of inversion

models (20 in this work) using different geothermal gradients, ranging from 50% to 120% of the
modern value, together with the *a priori* erosion rate estimated in the first step, for determining
the initial geothermal gradient that yields the maximum fit to the modern value, i.e., the minimum
$\Phi_\gamma$ (equation 13).

(iii) Optimizing the fit to both the age data and the geothermal gradient. Given the model

dependence on the geothermal parameters (see section 6.1), a comprehensive evaluation of the
models should assess not only the age misfit ($\Phi_\tau$), but also that of the geothermal gradient ($\Phi_\gamma$). In
the third step, a set of inversion models (20 in this work) are run using different *a priori* erosion
rates, changing from 10% to 200% of the mean value estimated in the first step, together with the
estimated geothermal gradient by the second step, to search for the model that provides the best fit
to both the age data and the modern geothermal gradient. This study uses the following compound
misfit function to evaluate the models:
$$\Phi = \Phi_\tau + \Phi_\gamma / \sqrt{N}, \qquad (17)$$
where $\Phi_\tau$ and $\Phi_\gamma$ are misfit values for the age and geothermal gradient calculated using the
equations 12 and 13, and $N$ is the number of age inputs. Dividing $\Phi_\gamma$ by the square root of $N$ in this
equation, as also done for calculating the $\Phi_\tau$ (equation 12), means that the modern geothermal
gradient is given the same weight as an age input for evaluating the model.

**8. Synthetic models for testing the new modeling guideline**
We firstly test our stepwise inversion scheme by synthetic datasets generated by thermo-
kinematic models modified from Braun et al. (2012) (their Fig. 9). The synthetic age dataset is
produced by *Pecube* using the following parameters: a steady-state topography with a 20-km
wavelength and a 2-km relief, a model block thickness of 30 km with a basal temperature of 600 °C,
a thermal diffusivity of 25 km$^2$/Myr, a sea level temperature of 10 °C, a lapse rate of 5 °C/km.
Worth noting is that these parameters are the same as Braun et al. (2012). For model details, see
Braun et al. (2012). For model setup see the supplementary Figure S1.
Synthetic AFT and AHe ages (supplementary Tables T1) were calculated for both surface
and borehole samples for four different exhumation histories. The synthetic models a and b are
characterized by a sudden decrease in exhumation rate from 1 km/Myr to 0.1 km/Myr (model-a,
same as the that shown in the Fig. 9 of Braun et al. 2012) and 0.3 km/Myr (model-b) at 5 Ma,
respectively. The models c and d include a sudden increase in exhumation rate from 0.3 km/Myr
(model-c) and 0.1 km/Myr (model-d) to 1 km/Myr at 5 Ma, respectively. All models start from 40
Ma. Except for the synthetic age data (plotted in the first row of Fig. 9), these four models generate
modern geothermal gradients of 26.5 °C/km, 28.6 °C/km, 35.5 °C/km and 34 °C/km for the
uppermost 2-km crust, respectively.

Inversion of rock exhumation history used a start time of 20 Ma and a time interval length

of 1.0 Myr for all synthetic datasets, which were assigned with a 6% uncertainty. As shown by the
modelling output visualized in Fig. 9a, our inversion of the rock exhumation from the synthetic
dataset-a finds an optimal initial geothermal gradient of 22 °C/km and *a priori* rate of 0.85 ± 0.25
km/Myr, and yields a decrease in exhumation rates from ~0.9 km/Myr (before 6 Ma) to 0.3-0.1
km/Myr (4-0 Ma), via a gradual decrease during 6-4 Ma. The data has no resolution for the
exhumation history before 10 Ma.  Comparing to the synthetic model (abrupt decrease from 1
km/Myr to 0.1 km/Myr at 5 Ma), the rates before 5 Ma are underestimated by 0.1 km/Myr, whereas
the values after 5 Ma overestimated by 0.1-0.3 km/Myr.

The inversion for the synthetic dataset-b results in an optimal initial geothermal gradient

of 21.7 °C/km and *a priori* rate of 0.81 ± 0.24 km/Myr, and an increase in exhumation rates from
~0.85 (before 5 Ma) km/Myr to 0.4-0.5 km/Myr (4-0 Ma), via a gradual decrease during 5-4 Ma
(Fig. 9b). Comparing to the synthetic model (abrupt decrease from 1 km/Myr to 0.3 km/Myr at 5
Ma), the rates before 5 Ma are underestimated, whereas the values before 5 Ma are overestimated
by ~0.1-0.2 km/Myr.

The inversion for the synthetic dataset-c yields an optimal initial geothermal gradient of

24.3 °C/km and *a priori* rate of 0.55 ± 0.17 km/Myr, and a decrease in exhumation rates from
~0.45-0.3 km/Myr (before 5 Ma) to 1.0 km/Myr (3-0 Ma), via a gradual increase during 5-3 Ma
(Fig. 9c). Comparing to the synthetic model (abrupt decrease from 0.3 km/Myr to 1.0 km/Myr at
5 Ma), the rates during 5-3 Ma are underestimated, whereas the rates before 5 Ma overestimated
by 0-0.15 km/Myr.

The inversion for the synthetic dataset-d produces an optimal initial geothermal gradient

of 24.5 °C/km and *a priori* rate of 0.25 ± 0.08 km/Myr, and an increase in exhumation rates from
~0.1-0.2 km/Myr (before 5 Ma) to 1.0 km/Myr (3-0 Ma), via a gradual decrease during 5-3 Ma
(Fig. 9d). Comparing to the synthetic model (abrupt decrease from 1 km/Myr to 0.3 km/Myr at 5
Ma), the rates before 5 Ma are slightly overestimated, whereas the values during 5-3 Ma are
underestimated.

To summarize, the inverted rock exhumation histories for the four synthetic datasets match

the whole picture of the synthetic "truth", but the variations in exhumation are underestimated,
and the sharp changes at 5 Ma are smoothed. It is worth noting that inversions using only surface
samples produce similar results (supplementary Fig. S2).

**9. Natural examples for testing the new modeling guideline**

Below we use three examples to demonstrate our new method. The Denali data is used

again for demonstrating the efficiency of our method in finding the proper initial geothermal
gradient and the *a priori* exhumation rate. Then, we further test our method using the Himalayan
Dhanladar range and KTB borehole (the Continental Deep Drilling Project in Germany)
thermochronologic data for representing regions of fast and slow erosion, respectively.

9.1 The Denali transect

Using the stepwise inversion modeling guideline, the Denali transect yields an exhumation

history generally similar with that of the reference model (Fig. 10a). Differences in the *a priori*
parameters include that the new inversion finds and uses an initial geothermal gradient of
25.2 °C/km (slightly higher than that of the reference model), *a priori* erosion rate of 0.46 ± 0.23
km/Myr (slightly lower than that of the reference model) and a time interval length of 1.5 Ma. The
combination of these *a priori* parameters result in a major increase in erosion rate to 0.55-0.6
km/Myr at 6 Ma, which is 1.5 Myr latter than that of the reference model (7.5 Ma). The subtle
differences from the reference model mainly result from the time interval length used in these
models. Comparing the misfit values, the new model produces slightly better fits than the reference
model, with the *a posterior* misfit values of 1.81 and 0.11 for the observed age and geothermal
data.

9.2 Himalayan Dharladar range transect
AFT and ZHe data from the Dharladar range in the northwestern Himalayas, reported in
the publications by Deeken et al. (2011) and Thiede et al. (2017) are used as an example for regions
of young cooling ages and fast exhumation. The samples were collected in an elevation range
between 1.5 and 4.5 km, covering a topographic relief of 3 km within a spatial distance of ~15 km
on the hanging wall of the main central thrust of the Himalayan fold-thrust-belt (Deeken et al.,
2011; Thiede et al., 2017). AER slope regression of ZHe and AFT ages performed in Deeken et al.
(2011) produced apparent erosion rates of ~2.8 km/Myr and ~0.2 km/Myr for the time intervals
6.4–14.5 Ma and 1.7–3.7 Ma, respectively, implying a potential increase in erosion rates at ~3.7-
6.4 Ma. Using geothermal gradients of 25-45 °C/km, time-averaged erosion rates were estimated
as 0.8-2.0 km/Myr since 3.7 Ma (Deeken et al., 2011).
The modelling of the Dharladar range data uses a modern geothermal gradient constraint
of 45 ± 8 °C/km (Deeken et al., 2011). The relatively large uncertainty is assigned for the
geothermal gradient, because of the absence of direct geothermal measurements in the study area.
Our exhumation inversion for the AER data using the stepwise modeling guideline yields relatively
slow rates of 0.1-0.6 km/Myr and fast rates of 1.2-1.6 km/Myr before and after ~3 Ma, respectively
(Fig. 10b). The abrupt increase of exhumation rates at ~3 Ma is generally consistent with the
estimates from the slope regression results of Deeken et al. (2011). However, the inverted
exhumation rates since 3 Ma are significantly lower than the estimation from the AER slope (~2.8
km/Myr), which is likely due to the overestimation of exhumation of the AER slope due to
topographic perturbation of isotherms. Such a perturbation is a function of exhumation rates: the
higher the exhumation, the larger the perturbation (Glotzbach et al., 2015). The modelling yields
a history of the geothermal gradient that gradually increases to a modern value of ~46 °C/km, close
to the input value (45 ± 8 °C/km).

9.3 KTB borehole
The KTB borehole yields a large thermochornologic and geochronologic age data
(Warnock and Zeitler, 1998; Stockli and Farley, 2004). Previous studies suggest the borehole are
truncated by multiple faults, which offset the age-depth relationship (Wagner et al., 1997). Here
we use the data at depths shallower than 1 km, where data are abundant and have linear relationship
with depths.
The KTB apatite, zircon and titanite (U-Th)/He (AHe, ZHe and THe) and AFT age data
vary largely between 85-50 Ma. These clustered ages have been interpreted as indicating a late
Cretaceous phase of exhumation, followed by slow exhumation (Wagner et al., 1997; Stockli and
Farley, 2004), as also shown by previous thermal history reconstructions based on k-feldspar
[40]Ar/[39]Ar data (Warnock and Zeitler, 1998).
Our modeling, using the AER data and a modern geothermal gradient of 27.5 ± 2.8 °C/km
(Clauser et al., 1997), shows that elevated exhumation rates (0.1-0.13 km/Myr) between 80-50 Ma,
followed by slower exhumation rates of ~0.04 km/Myr (Fig. 10c), are similar to previous estimates
(Wagner et al., 1997; Warnock and Zeitler, 1998; Stockli and Farley, 2004). Associated with
changes in exhumation, geothermal gradient gradually decreases from the peak values at 70-60
Ma to a value of ~28 °C/km at the present-day.

**10. Conclusion**
The *a priori* information has important effects on the inversion results using the least-
squares inversion method. Our study demonstrates the importance of geothermal gradient and the
*a priori* exhumation rate in estimating the exhumation history from the thermochronology data.
To take into account the geothermal data into the exhumation history inversion, we outlined a
stepwise inversion method that first searches for the appropriate initial geothermal gradient, which
is then used in the modelling searching for the *a priori* exhumation rate. Our modelling guideline
produces exhumation history and geothermal gradient that provide reasonable fits for both the
observed AER and modern geothermal data, as tested by datasets of both synthetic models and
natural samples. The code and data used in this work are available in GITHUB
(https://github.com/yuntao-github/A2E_app).

**Code availability**
The code used in this work are available in GITHUB (https://github.com/yuntao-github/A2E_app).

**Data availability**
The data used in this work are available in GITHUB (https://github.com/yuntao-github/A2E_app).

**Author contribution**
Yuntao Tian: Conceptualization, Methodology, Software, Data curation, Visualization,
Investigation, Writing- Original draft preparation. Lili Pan: Visualization, Writing- Reviewing and
Editing. Guihong Zhang and Xinbo Yao: Writing- Reviewing and Editing.

**Competing interests**
The contact author has declared that none of the authors has any competing interests.

**Acknowledgments**
This study is funded by the National Natural Science Foundation of China (42172229, 41888101
and 41772211). Discussions with Jie Hu and Donglan Zeng are gratefully appreciated. Comments
and suggestions from Gilby Jepson and Christoph Glotzbach clarified many points of this work.

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

**Figures captions:**

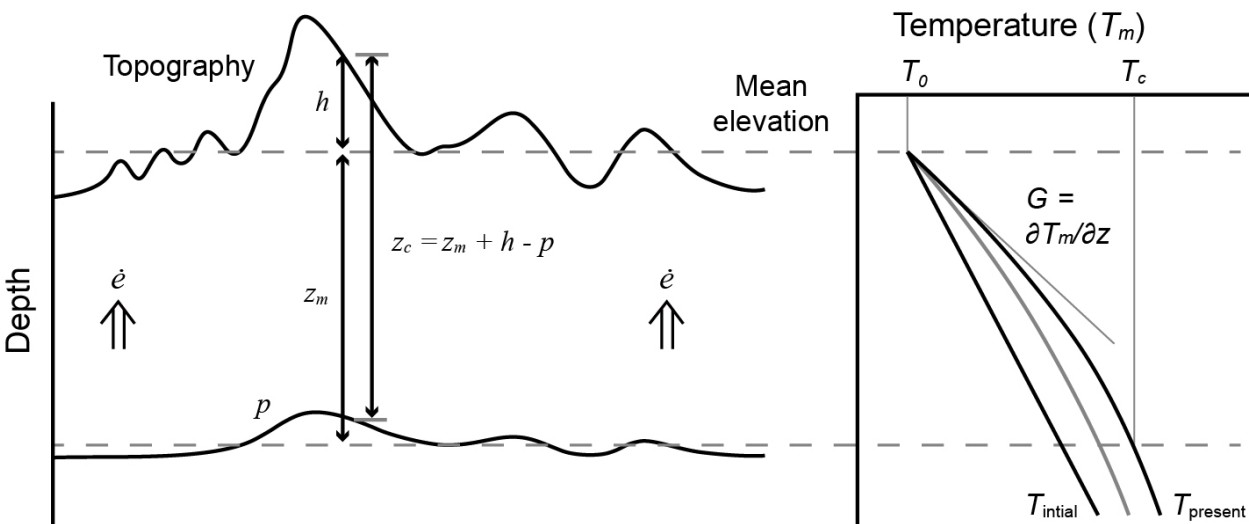


Figure 1. Schematics showing the relationship among closure depth ($z_c$), topography and its
perturbation (p). The parameter $h$ denotes the difference between the sample and the mean
elevation, and $z_m$ the depth of the closure temperature ($T_c$, the lower dashed line) derived from
the mean elevation (upper dashed line) and intial temperature field ($T_{initial}$) and exhumation
history ($\dot{e}$).


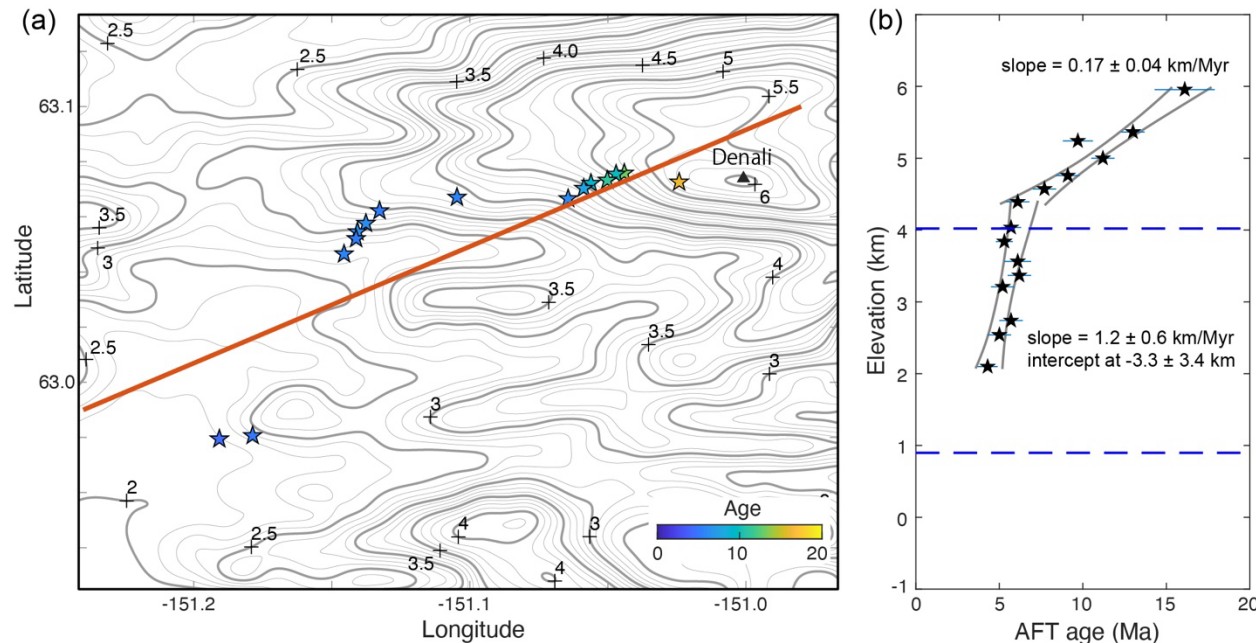


Figure 2. (a) Distribution of AFT age data (pentagons, colored by age values) over the elevation

contour map computed using the SRTM30 data of the Denali massif in Alaska. AFT data

sourced from Fitzgerald et al. (1995). (b) AER and the slope fitting results using isoplotR

(Vermeesch, 2018). AER fitting of ages older than 6.7 Ma yields a slope of $0.17 \pm 0.04$ km/Myr;

whereas the fitting of ages between 6.5 Ma and 4.3 Ma produces a slope of $1.2 \pm 0.6$ km/Myr

and an intercept at -3.3 ± 3.4 km. The upper and lower dashed lines denote the mean elevation

(4.02 km) and the depth of the nominal closure temperature (110 °C), calculated using the

modern geothermal gradient (38.9 °C/km) and the surface temperature (-12 °C).


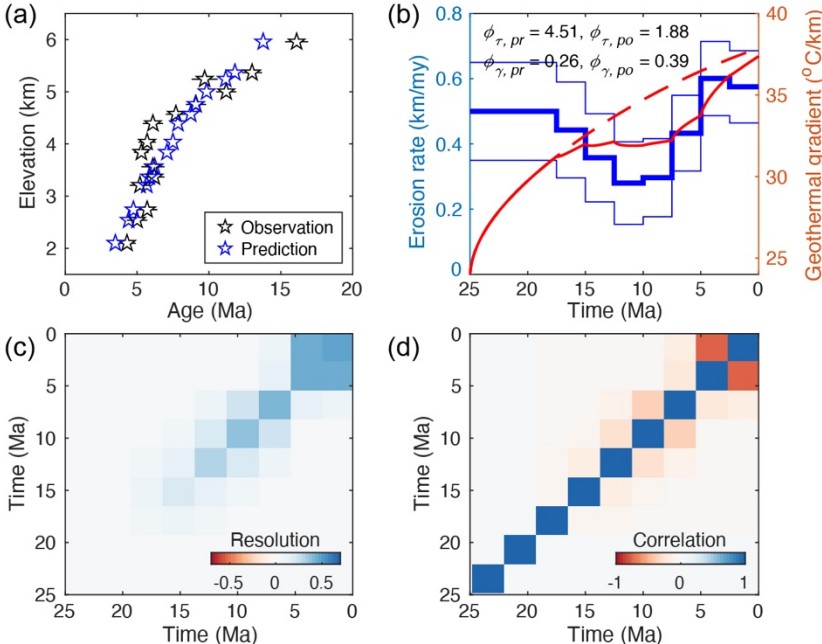


Figure 3. Inputs and outputs of the reference model for the Denali AFT. (a) Comparison between
the observed (in black) and predicted (in blue) AER. (b) The *a posterior* exhumation history
generated by the reference model. Thick and thin lines are the mean and one standard deviation
of the inverted exhumation history. The red dash and solid lines are the history of the geothermal
gradients, predicted by the *a priori* and *a posterior* models, respectively. (c) and (d) Plots of the
resolution and correlation matrix.

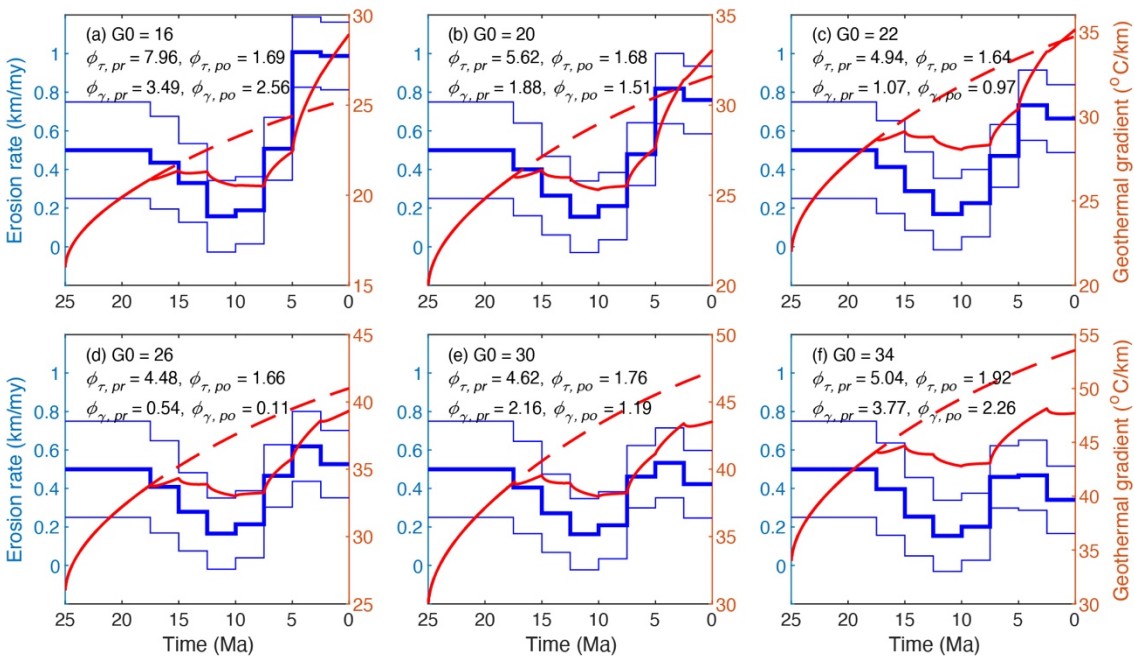


Figure 4. Histories of exhumation and geothermal gradients, predicted by models using different
initial geothermal gradients between 18 °C/km and 34 °C/km. The blue thick and thin lines are
the mean and one standard deviation of the inverted exhumation history. The red dash and solid
lines are the history of the geothermal gradients, predicted by the *a priori* and *a posterior*
models, respectively. Except for the initial geothermal gradient, other parameters are the same as
the reference model. Comparing to the reference model which used an initial geothermal gradient
of 24 °C/km (Fig. 3), models using a lower initial geothermal gradient yield relatively higher
exhumation rates (panels a-c), whereas those using a higher gradient produce lower exhumation
rates (panels d-f).

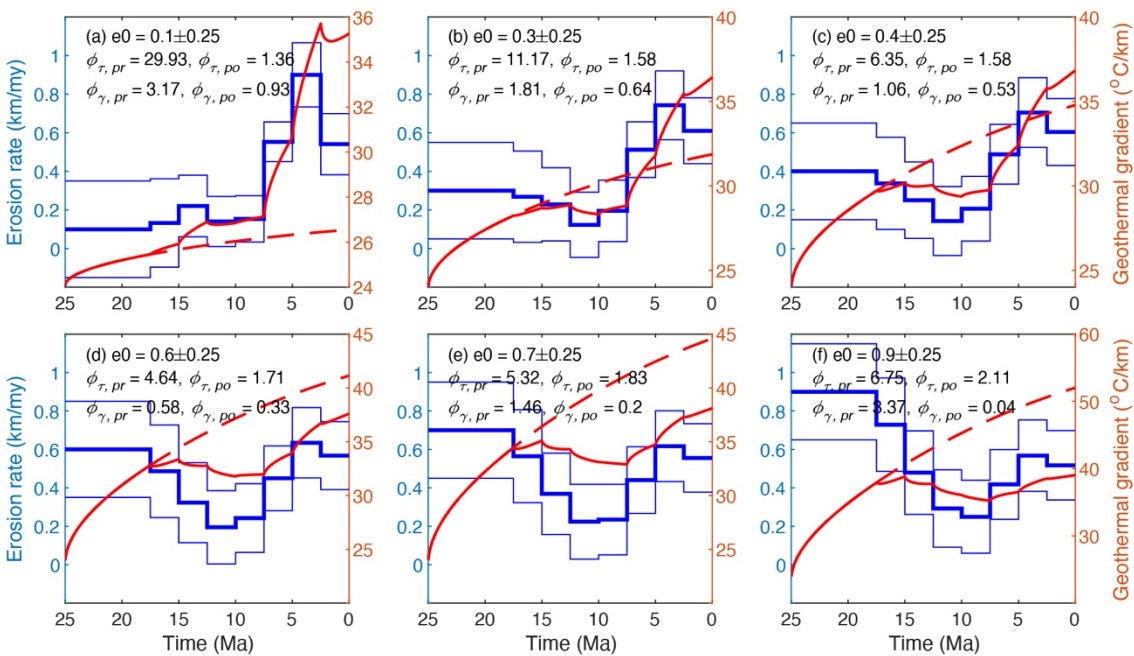

Figure 5. Histories of exhumation and geothermal gradients, predicted by models using different *a priori* mean values of the exhumation rates, ranging from 0.1 km/Myr to 0.9 km/Myr. Other parameters are the same as the reference model. For explanation of the plotted lines, see Figure 4. Comparing to the reference model which used *a priori* mean exhumation of 0.5 km/Myr (Fig. 3), models using a lower *a priori* exhumation yield relatively higher exhumation rates for the last three stages (7.5 - 0 Ma) (panels a-c), whereas those using a higher *a priori* exhumation produce lower exhumation rates for the last three stages (panels d-f).

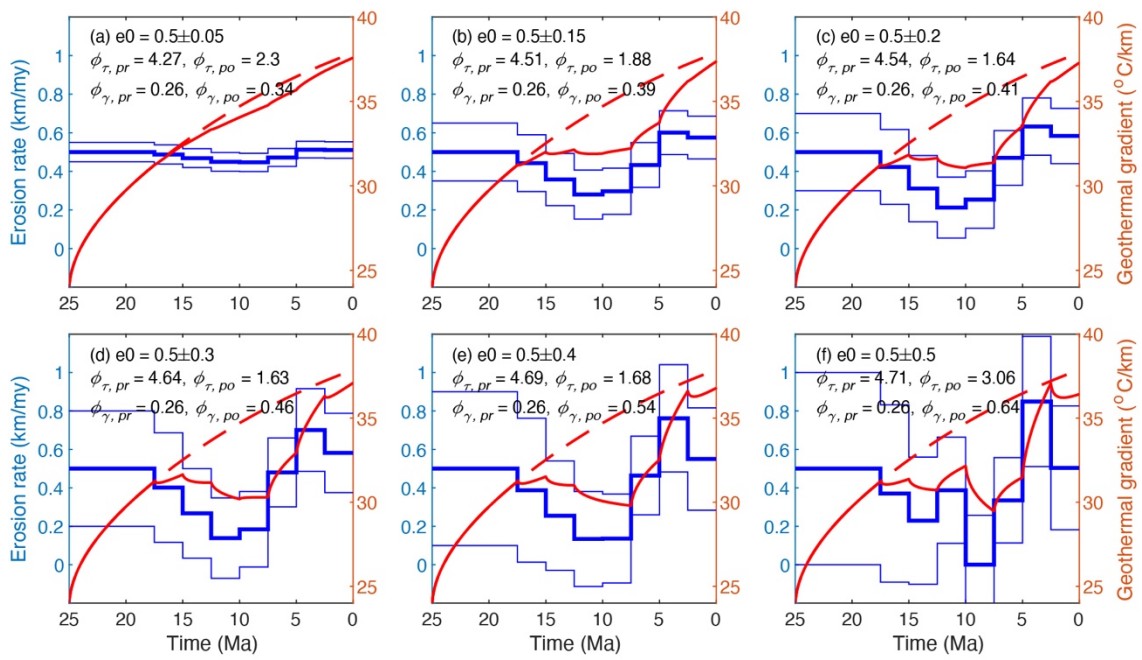


Figure 6. Histories of exhumation and geothermal gradients, predicted by models using different
*a priori* variance values (between 0.05 km/Myr and 0.5 km/Myr) of the exhumation rates (0.5
km/ Myr).  Other parameters are the same as the reference model. For explanation of the plotted
lines, see Figure 4. Comparing to the reference model which used *a priori* variance of the
exhumation (0.25 km/Myr) (Fig. 3), models using a lower *a priori* variance yield limited
variations and uncertainties in exhumation (panels a-c), whereas those using a higher *a priori*
variance produce larger variations and uncertainties (panels d-f).

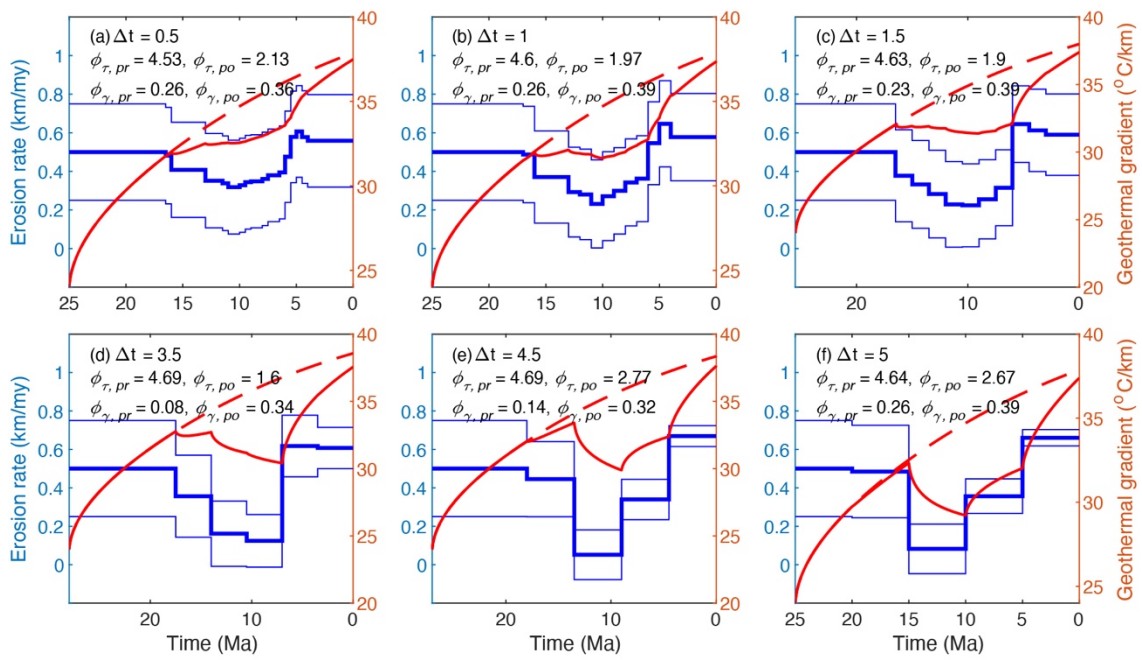


Figure 7. Histories of exhumation and geothermal gradients, predicted by models using different
time interval lengths. Other parameters are the same as the reference model. For explanation of
the plotted lines, see Figure 4. Comparing to the reference model which used a time interval
length of 2.5 Ma (Fig. 3), models using smaller time interval lengths yield lower variations in
exhumation (panels a-c) than other using larger time interval lengths (panels d-f).

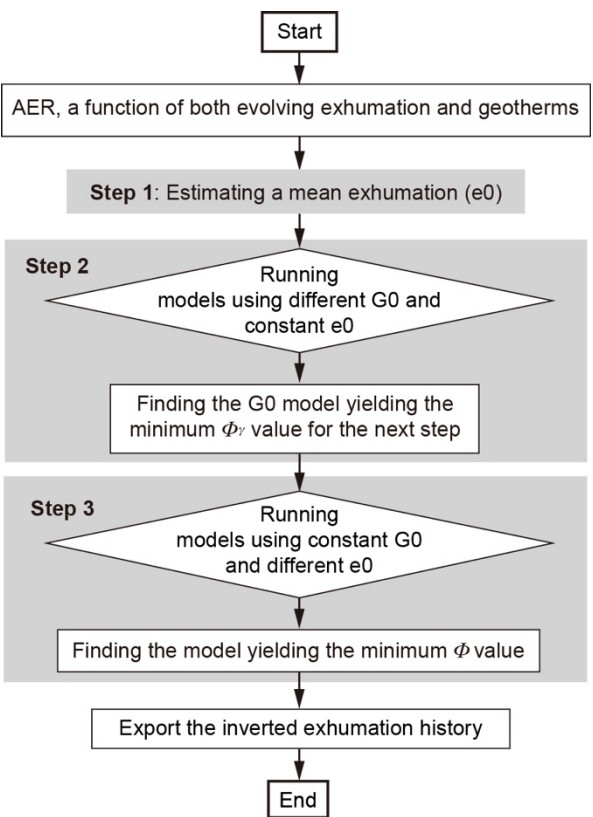


Figure 8. Flow chat of a stepwise modeling method, which includes three main steps. The first
step estimates a mean exhumation rate (e0) using the nominal closure temperatures, modern
geothermal gradient and sample ages. The mean rate is used in the second step which runs a set
of models using different initial geothermal gradients for optimizing the initial geothermal
model. The third step runs a set of models using different *a priori* exhumation rates, which is
generated around the mean rate, and the optimized initial geothermal model by the second step,
to find the best model that yields the minimum misfit to both age data and modern geothermal
gradient.

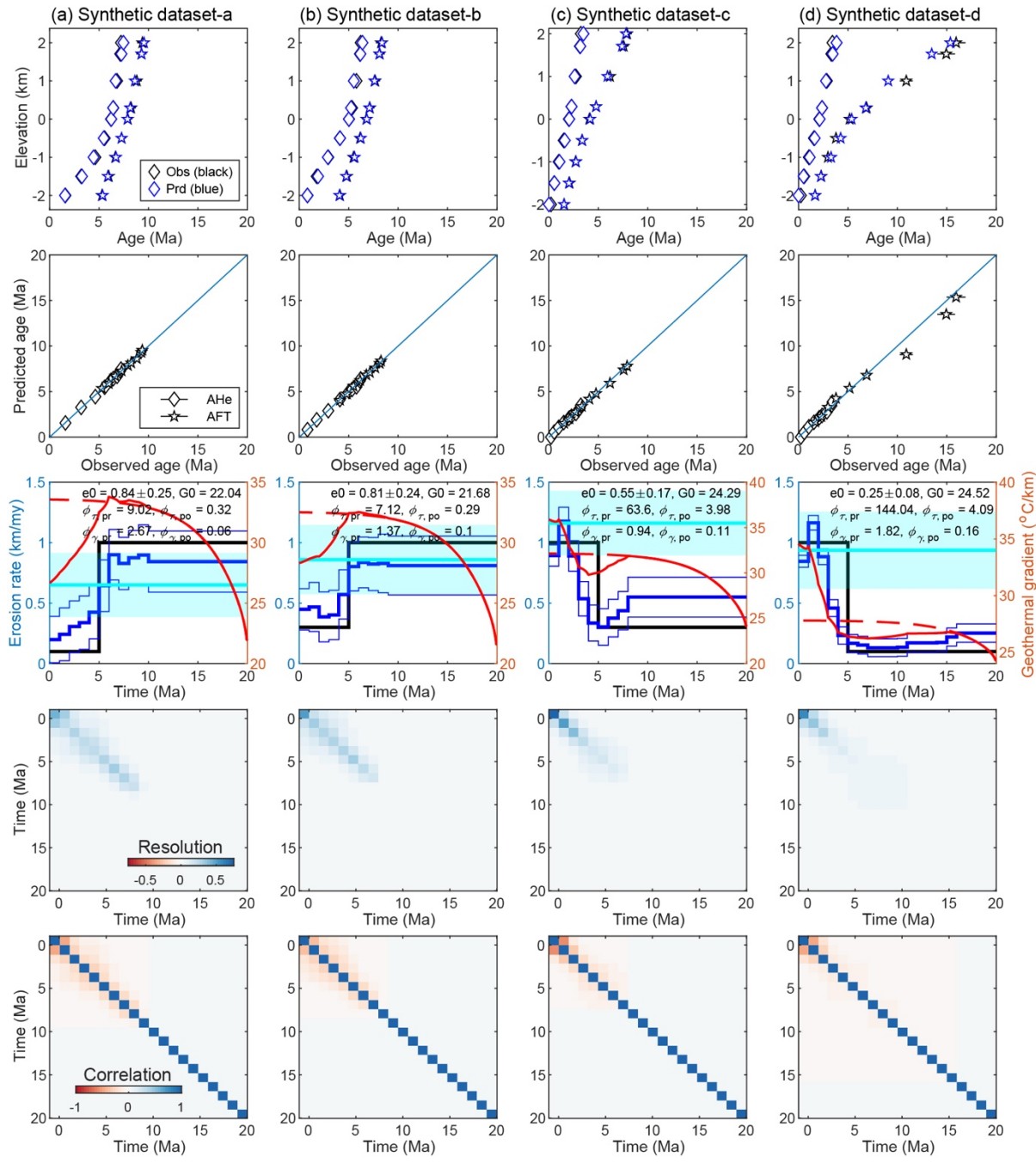


Figure 9. The best-fit model for the synthetic dataset-a, -b, -c and -d using the modeling method

shown in figure 8. First row: Comparison between the observed (in black) and predicted (in blue)

AER. Second row: plots of observed and modeled ages. Third row: Histories of exhumation and

geothermal gradients. The black line marks the "true" exhumation history used for simulating the

age dataset, whereas the blue thick and thin lines are the mean and one standard deviation of the

inverted exhumation. The red dash and solid lines are the history of the geothermal gradients,
predicted by the *a priori* and *a posterior* models, respectively, whereas the cyan line and polygon
denotes the modern geothermal gradient. Fourth and bottom row: Plots of the resolution and
correlation matrix.

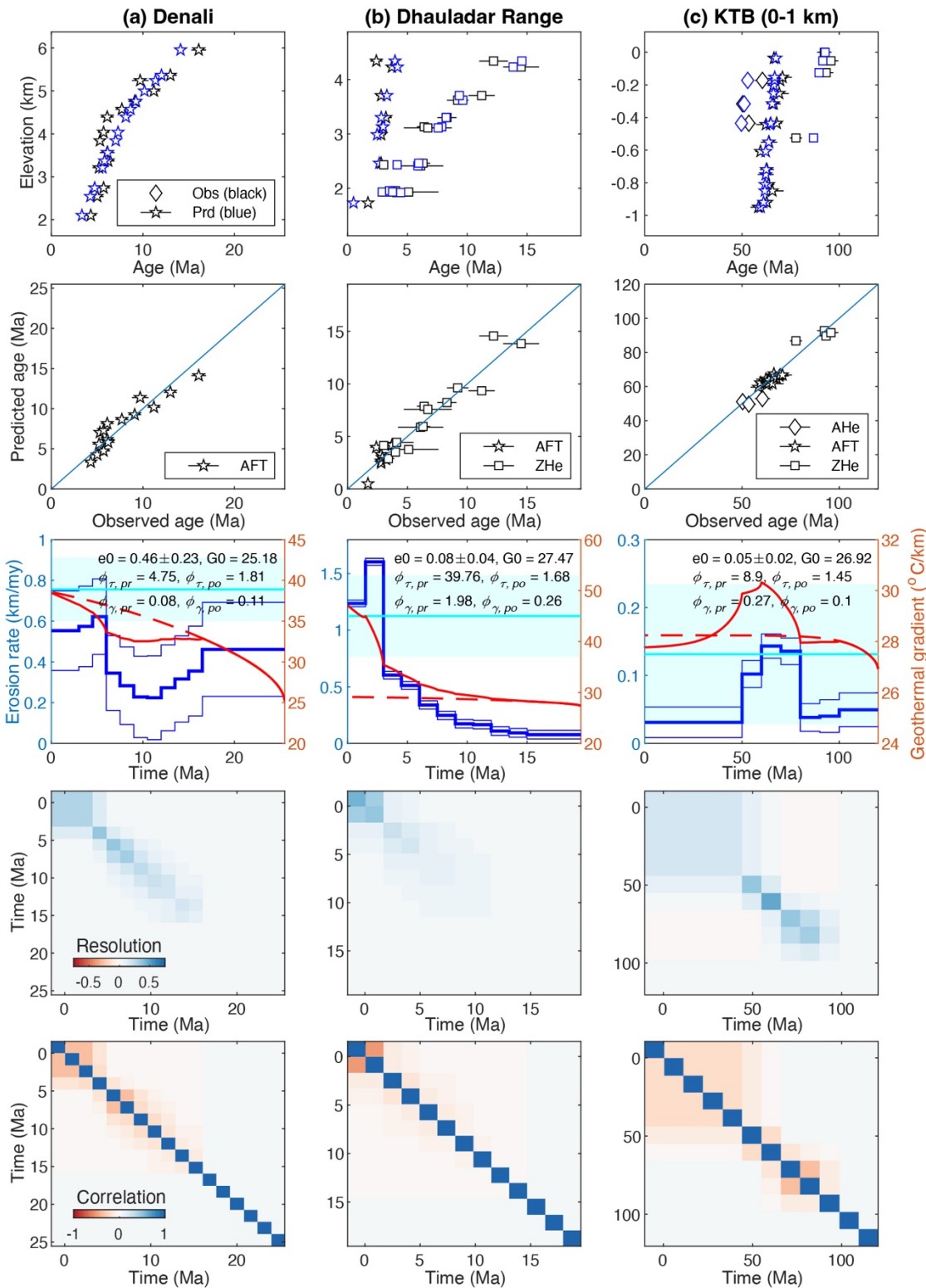


Figure 10. The best-fit model for the Denali (a), Dhanladar range (b) and upper KTB (c)

transects, using the modeling method shown in figure 8. See Fig. 8 for panel interpretations.