# Peer review of "An efficient approach for inverting rock exhumation from thermochronologic age-elevation"

_EGUsphere, 2023_

## Author Comment (AC1)

Many thanks for your comments and suggestions, which have been very helpful for further improving the clarity of our manuscript.

Below please find a detailed response to the comments and suggestions received, and a summary of how the manuscript has been revised. The original comments are copied here in black text, followed by our responses in blue.

In "An efficient approach for inverting rock exhumation from thermochronologic age-elevation relationship", Tian and co-authors present a least-squares inversion method for solving exhumation history from thermochronologic age-elevation relationships. Their approach seems sound identifying that the a priori erosion rate, time-interval, and geothermal gradient are all important controls on erosion rates, similar to those that have come before (e.g., Braun, 2003, van der Beek et al., 2010 or Fox et al., 2014). Certainly, it is suitable for ESurf. However, I have some queries regarding how the model handles certain parameters.

Dependence on time interval length: The authors spend some time discussing the suitability to time step interval. This makes sense. With larger steps, one can average into larger changes in erosion rates. I am wondering if selecting a suitable interval would not this be a function of the uncertainty of the underlying chronometer? E.g., if using largely systems with high uncertainty (e.g., AFT) then only a larger bin-size should be used. Whereas systems with higher precision (Ar-Ar, He) can resolve finer variations?

Responses: This is the one of the shortcomings of the linear method, which solves the age-exhumation-closure depth equation with a specific time interval length. Hopes future studies can address this issue.

Testing examples: I am curious as to why the background erosion rate for the Dhauladar range could be considered so low for so long? Okay, 0.2-0.4 mm/yr is maybe not "low" in other orogenic systems. However, this is the Himalaya. Additionally, in Deeken et al., 2011, they are considering that the region has experienced high erosion rates for an extremely long time, facilitating channel flow. Did the authors consider a high a priori erosion rate in this system? Regardless, given the importance of the prior erosion rate to the predicted erosion rate, I believe the authors should discuss possible approaches.

Responses: We did run a set of models using different a priori erosion rates. As stated in the section '7. A new modeling strategy', the third step of the new model strategy runs "… a set of inversion models (20 in this work) are run using different *a priori* erosion rates, changing from 20% to 150% of the mean value estimated in the first step, together with the estimated geothermal gradient by the second step, to search for the model that provides the best fit to both the age data and the modern geothermal gradient …"

Below is a screen shot of the plots every four models with different prior erosion rates.

The model presented in the paper provides the best fit to both the age data and the modern geothermal gradient, i.e., the lower compound misfit (equation 7).

The apparent erosion rates calculated from the AER slope of ZHe data is indeed low (0.2 mm/yr). See the second column of the Table 2 of Deeken et al. (2011).

[Figure]

The KTB borehole seems to be most poorly performing of the 3 tests. Not horribly, but something to notice. Specifically, the model seemly predicting ages that are systematically younger than their observed ages. Particularly, getting into the lower T chronometers (AHe and AFT). Could this be a function of performing this study in a borehole where temperatures do increase with depth? Further, what is the a priori erosion rate and variance?

Responses: Thanks to this comment, we identified a bug for calculating the posterior AHe age prediction function. It is fixed.

The relatively worse fit to the AHe ages results from the following two factors. (1) the relationship between depth and age is non-linear, which makes to impossible to fit the AHe age at a depth of -0.4 km. (2) The uppermost AHe age (at a depth of -0.2 km) is almost identical (or even a bit older than the AFT ages at a depth of -0.9 km). Considering the significantly lower closure temperature of the AHe method and the slow exhumation of the region, it is also impossible to fit the uppermost AHe age.

Detailed comments:

Lines 20-22: "Modelling experiments demonstrate the significant and systematic influence of initial geothermal model, the a priori exhumation rate and the time interval length on the a posterior exhumation history". Here the authors state this correlation in the abstract, yet in their demonstrated examples, there is little discussion on these topics.

Responses: the section '6. Dependence on model parameters and proposed solutions' aimed to demonstrate the influence of initial input parameters on the exhumation history.

Lines 41-42: "..ranging from mountain building (e.g., Zeitler et al., 2001; Whipp Jr. et al., 2007; Cao et al., 2022) and its decay (e.g., House et al., 2001..". suggestion: "orogenic growth and decay".

Responses: revised.

Lines 51-52: "such as mica Ar-Ar, apatite, zircon and titanite fission-track and (U-Th)/He analyses..". Given the wide variety of accessory phases that these decay systems are found in, I would simply omit all the possible mineral examples. As there is 3 mineral phases for FT, none for U-Th and one for Ar-Ar.

 Responses: revised.

Lines 64-65: "patiotemporal changes in exhumation (Sutherland et al., 2009; Herman et al., 2013; Fox et al., 2014; Willett et al., 2020)". Please include: "van der Beek, P. and Schildgen, T. F.: Short Communication: age2exhume – A Matlab script to calculate steady-state vertical exhumation rates from thermochronologic ages in regional datasets and application to the Himalaya, EGUsphere [preprint], https://doi.org/10.5194/egusphere-2022-888, 2022."

Responses: The suggested reference is added, and categorized as a method for determining 'time-averaged exhumation rates' for each thermochronometric age.

Line 75: "Because both the underground geothermal field..". suggestion: "As both the subsurface geothermal field..". Also, the authors use "underground" a bit. I think "subsurface" might be the better descriptor.

Responses: revised.

Line 78-80: "reliable estimates of exhumation rates require solving exhumation itself, together with the evolution of other influencing factors". This sentence needs some clarification it is difficult to see what the authors are driving at here.

Responses: To clarify this statement, it is revised as "… reliable estimates of exhumation rates require solving exhumation itself, together with the evolving geothermal field and the closure temperatures".

Line 200: "Same as used in Fox et al. (2014)..". Suggestion: "following protocol outlined in".

Responses: revised.

Line 206: "..results reveal an abrupt triple-four-fold increase of exhumation rate..". a "tripe-four-fold" seems a little confusing. Just pick either 3-fold or 4-fold.

Responses: revised.

Lines 251-255: "Worth noting is that 252 the models using relatively lower (16-20 C/km, Figs. 4a-4b) and higher (30-34 oC/km, Figs. 4e-4f) initial geothermal gradients yield relatively worse misfits (>1) than those using medium initial gradients (22-26 C/km) (Figs. 3 and 4c-4d), suggesting that the modern geothermal gradient can be used as a constraint for the initial geothermal model. In the above section the authors suggest that the present day geothermal gradient is ~40 C/km (line 203). Given that the authors find lower misfit in the "medium initial gradients", how does that fit with the claim "modern geothermal gradient can be used as a constraint for the initial geothermal gradient?

Responses: The models presented in Fig 4 and explained in the section '**6.1. Dependence on initial thermal model' use different initial geothermal gradient. Each of these models predict a** present-day geothermal gradient, which is compared with the observed value (39.8 C/km) to determine the misfit between them. As shown the models use medium initial gradients (22-26 C/km) predict present-day geothermal gradients that are similar to the observed value (39.8 C/km). Therefore, "the observed modern geothermal gradient can be used as a constraint for the initial geothermal gradient".
The statements are slightly reworded to make it clearer. "…Further, it is also shown that models using higher and lower prior geothermal gradients of <20 °C/km (Figs. 4a-4b) and >30 °C/km (Figs. 4e-4f) yield worse misfits ($\Phi_{\gamma, po} > 1$) for the observed present-day geothermal gradient than those ($\Phi_{\gamma, po} < 1$) using medium initial gradients (22-26 °C/km) (Figs. 3 and 4c-4d). …"

Line 284: "We proposed to run a set of models..". Here and throughout, the authors use "proposed" however, it is unclear what they mean by 'proposed'. In my view, if one is proposing to do something, they crucially, have not done it. However, the authors have, presumably, done some of the things they are proposing. Thus, I would just have a think if that is specifically the word they would like to use.

Responses: Partly revised. We kept using the term 'proposed' in the section 6, where we presented a set of modeling experiments to demonstrate the dependence of modeling output on the input parameters, and suggested ("proposed" as used in the manuscript) potential ways to address the dependence.

In the following section 7, we outlined a stepwise modeling method to implement the "proposed" modeling guidelines.

**Therefore, in the revised manuscript,** the term 'proposed' in the section 6, and the usage in other sections are replaced by 'outlined', or 'implemented'.

Lines 286-287: "As to the a priori variance of erosion rates, we propose to use a relative uncertainty of 30-70% of the mean value." Here I am confused. Do the authors propose that users should set relative uncertainty to 30-70% or are they say that they set relative uncertainty to this value within their models? Given that the authors discuss variance. Is there a mean value that gives better fits? It is likely system dependent, but perhaps making a statement is useful here.

Responses: The a priori variance is one of the inputs for the modeling, i.e., the $_{pr}$ in the equation 2. We propose to use a value 30-70% of the a priori erosion rates.

These lines are revised to clarify this point.

As to the comment "Is there a mean value that gives better fits? It is likely system dependent, but perhaps making a statement is useful here." We found that "Further, models using the input variance of the *a priori* exhumation of 0.2-0.3 km/Myr (40-60% of the mean value), the variation of the inverted exhumation history becomes stable (Figs. 3, 6c-6d)" (lines 279-280 of the original manuscript). We agree that it is likely to be dependent on input thermochronological data. Following this suggestion, we added a statement "Future applications of the method may need to test a set of the variance inputs so as to get a stable exhumation output.".

Line 293: "Using a large time length..". missing "interval".

Responses: revised.

Line 314: "these aren't really "new modelling strategies", as many of the previously mentioned modelling systems involve the same parameters. Thus, I would suggest making this more specific. modelling guidelines?

Responses: revised by replacing the term 'strategy' by 'guideline' or 'method'.

Line 315: "Putting together the lessons learned from the..". Following the modelling protocol outlined above.

Responses: revised.

Line 316: "modeling strategy develops..". is developed.

Responses: revised.

Line 361: "..Dharladar range in the central Himalayas..". northwestern himalaya?

Responses: revised.

Line 366-367: "AER slope regression suggests an increase in apparent erosion rates from ~0.2 km/Myr to ~2.8 km/Myr at ~3.7-6.4 Ma (Deeken et al., 2011)." Looking at the Deeken et al., ageelevation plots, the inflection point seems a little ambiguous. Does your approach document significant change if you use the "no clear break in slope' approach?

Responses: This statement is based on the Table 2 of Deeken et al., 2011. As shown in that table, regression of AFT ages produces an apparent erosion rate ~2.8 km/Myr during 1.7–3.7 Ma, whereas regression of ZHe yields a value of ~0.2 km/Myr during 6.4–14.5 Ma.

To clarify it, this statement is revised as below. "… AER slope regression of ZHe and AFT ages performed in Deeken et al. (2011) produce apparent erosion rates of ~2.8 km/Myr and ~0.2 km/Myr for the time intervals 6.4–14.5 Ma and 1.7–3.7 Ma, respectively, implying a potential increase in erosion rates at ~3.7-6.4 Ma."

Line 386: "The KTB apatite, zircon and titanite (U-Th)/He (AHe, ZHe and THe) and AFT age data vary largely between 85-50 Ma." Looking at figure 9, it seems that the models respond better to the less precise systems (AFT) then compared to the more precise systems of AHe. Do the authors also observe this? If so, why would this be the case?

Responses: Thanks to this comment, we identified a bug for calculating the posterior AHe age prediction function. It is fixed.

I thank the authors for their time.

Responses: We are deeply grateful for your reviews, which helped to clarify many points of our manuscript.

---

## Author Comment (AC2)

Many thanks for your comments and suggestions, which have been very helpful for further improving the clarity of our manuscript.

Below please find a detailed response to the comments and suggestions received, and a summary of how the manuscript has been revised. The original comments are copied here in black text, followed by our responses in blue.

The topic of the manuscript, estimating robust estimates of exhumation from thermochronological data, is an important method to study the spatial and temporal relations of tectonic, climate and erosion in various tectonic settings. Numerous papers have been published on this, ranging from simple geometric models, thermal history modelling to 1D to 3D thermal-kinematic models. The contribution of Yuntao Tian et al. is based on the 1D inversion of thermochronological data from age-elevation profiles sharing the same exhumation history. They do largely use the approach of Fox et al. (2014) with, as I understand, only very slight modifications of the data covariance matrix. It is important that this is stated clearly, and that the novelty of their contribution is easy to digest from the reader. It does largely read as a new method, but in most cases simply reprints equations/ideas formerly stated by Fox et al. (2014).

Response: This work followed the method of Fox et al. (2014). We aim to form an automatic and objective workflow that takes into account numerous prior input parameters (such as a priori exhumation rates and its variance, initial geothermal gradient, time interval length etc) to invert rock exhumation from age-elevation relationships (AERs) derived from both surface vertical transects and boreholes. Such a method is required for two reasons: (1) it provides an objective assessment on the models based on the misfits in both age and geothermal gradient. Note that the method of Fox et al. (2014) and its applications did not take the geothermal gradient into account in the model assessment, although the paper did highlight the importance of the parameter in influencing the inverted results. (2) Our new workflow enables researchers to handle a large number of AERs, which are available in many orogens.

Please see my scientific comments and technical corrections for more details:

1.  In the abstract, it is unclear what is the outcome of your model application. You state that 'lessons learned from the experiments', and afterwards, you propose a modelling strategy. One could get the impression that you have explored the difficulties but have not applied it.

    Response: In fact, we have implemented (rather than "propose") a new workflow. The statement has been revised as "Putting together these findings, we implemented a new stepwise inverse modeling method for optimizing the model parameters by comparing the observed and predicted thermochronologic data and modern geothermal gradient to mitigate the model dependencies on the initial parameters."

2.  Your approach is largely based on (a reprint of) that from Fox et al. (2014). Please make sure to clearly communicate that you are using his approach with only very slight modifications. It is often not directly recognizable if your equations are new or have been defined elsewhere. In fact, it would be easier to recognize your contribution if you put everything out of the manuscript (in the supplement) that is not new and just focus on your contribution and refer to Fox et al. (2014) for the method.

Response: This work followed the method of Fox et al. (2014). We clarified this in the beginning of the section 2. See above for our own contributions.

3. You applied your suggested modelling strategy to three examples and yielded 'consistent' results. This is good, but what is the benefit of running these models if you could just reproduce what others predicted before with 'simpler' methods? Please clearly state what is new and why we should use this modelling strategy and also what is different from what Fox et al. (2014) suggested?

   Response: Our results are not always consistent with previous results, for example the inversion of the himalayan Dharladar range transect. As for the KTB example, previous studies did not quantify the rock exhumation history, but cooling histories.

   We use this example to highlight that the new method can handle borehole samples.

4. It would be very nice to extend the applications to a synthetic data with changing exhumation rate and pronounced topography. In Braun et al. (2012) you will find two datasets (Fig. 3 and Fig. 9) that you could use to show the performance of your dataset in comparison to 1D thermal-kinematic modelling. Whatever the outcome is, you could discuss the limitations of some of your model parameterizations, such as the closure depth.

   Response: This suggestion is very useful. We run four synthetic models, please find the details in the section '8. Synthetic models' of the revised manuscript. It is shown that "…the inverted rock exhumation histories for the four synthetic datasets match the whole picture of the synthetic "truth", but the variations in exhumation are underestimated, and the sharp changes at 5 Ma are smoothed…"

**Technical corrections:**

Line 24: It is not clear what you mean with 'lower inversion results', please be specific.

Responses: revised as "a relatively higher a priori exhumation rate would lead to systematically lower a posteriori exhumation, and vice versa"

Line 40: Say something like 'Quantifying rock exhumation…'.

Responses: revised

Line 48: Give references to modelling tools such as Pecube (Braun 2003; Braun et al. 2012) and Glide (Fox et al. 2014).

Responses: revised

Line 53-58: Mention that this approach is only applicable for constant cooling rates.

Responses: revised

Line 59: Change to 'Many analytical …cooling history from thermochronological data.'

Responses: revised

Line 59-67: You may also want to mention the Fourier approach for correction AER from Glotzbach et al. (2015).

Responses: The Glotzbach et al. (2015) is cited and classified into one of the methods for inverting "time-averaged exhumation rates".

Line 78-80: Change to '…Ehlers and Farley, 2003). Estimating reliable exhumation rates requires to account for temporal variations of the thermal field caused by changes in the thermal and kinematic boundary conditions.'

Responses: revised

Line 86-87: I would delete the last sentence.

Responses: revised

Line 90-122: Do mention at the beginning that you are using the approach from Fox et al. (2014) with slight modifications, e.g. covariance.

Responses: Our inversion of exhumation from thermochronologic data followed the linear inversion approach of Fox et al. (2014).

Line 126-128: Replace with 'The latter can be determined modelling the temperature of the crust using a 1D thermal-kinematic model, which accounts for heat conduction, advection and production…'

Responses: revised

Line 149: Please give a reference for this equation, I guess Mancktelow and Grasemann (1997) and Fox et al. (2014).

Responses: revised

Line 169: Replace fitness with 'difference between observed and predicted ages weighted by the observed analytical uncertainty'.

Responses: revised

Line 176: Replace 'data fitness' with 'model result'.

Responses: revised

Line 190: Delete 'change'.

Responses: revised

Line 227: Give a reference for this equation.

Responses: revised

Line 251-255: Simplify and say that '…suing the present-day geothermal gradient (38.9 °C/km) in the misfit calculations does exclude higher and lower prior geothermal gradients of >30 and <20…'

Responses: revised

Line 355: What do you mean with 'latter'?

Responses: the statement is clarified as "…which is 1.5 Myr latter than that of the reference model (7.5 Ma)."

Line 375-376: Your estimate of the most recent exhumation is much lower compared to the raw interpretation, is that due to the overestimation of exhumation due to the topographic perturbation of isotherms?

Responses: Indeed, it is due to overestimation of exhumation due to the topographic perturbation of isotherms. We added the following statements. "AER slope regression of ZHe and AFT ages performed in Deeken et al. (2011) produced apparent erosion rates of ~2.8 km/Myr and ~0.2 km/Myr for the time intervals 6.4–14.5 Ma and 1.7–3.7 Ma, respectively, implying a potential increase in erosion rates at ~3.7-6.4 Ma..." … "… However, the inverted exhumation rates since 3 Ma are significantly lower than the estimation from the AER slope (~2.8 km/Myr), which is likely due to the overestimation of exhumation of the AER slope due to topographic perturbation of isotherms. Such a perturbation is a function of exhumation rates: the higher the exhumation, the larger the perturbation (Glotzbach et al., 2015)…"